# Solutions for SMEs Challenged by CSR: A Multiple Cases Approach in the Food Industry within the DACH-Region

**Angélique Catharina Elford** [1,*] **and Claus-Heinrich Daub** [2]

[1] School of Business, University of Applied Sciences and Arts Northwestern Switzerland, Riggenbachstrasse 16, 4600 Olten, Switzerland

[2] Institute of Management, University of Applied Sciences and Arts Northwestern Switzerland, Bahnhofstrasse 6, 5210 Windisch, Switzerland

\* Correspondence: angeliqueelford@yahoo.de

**Abstract:** Antecedent research has recognized a difference in the participation in Corporate Social Responsibility (CSR) practices between big companies and small- and medium-sized enterprises (SMEs). Certain characteristics of SMEs create challenges which influence the manner with which they treat the CSR topic. However, literature has failed to provide solutions as to how these challenges can successfully be overcome or avoided by SMEs. In an attempt to contribute to these solutions, this paper explores the reasons why some SMEs face challenges as well as how such problems can be mastered. Furthermore, this article provides input that could encourage more SMEs to incorporate CSR practices into their business strategies. The research follows a qualitative approach; data being collected in 2018 through a total of 12 interviews with managers of SMEs in the food industry within the DACH-region as well as with experts in the field of study. The paper reveals that if the managers and owners of SMEs become committed towards CSR and if sufficient resources are allocated and advice on how to implement CSR is obtained, the opportunity certainly exists to persuade larger numbers of SMEs to adopt CSR practices as a core company strategy.

**Keywords:** corporate social responsibility; strategies to sustainable development; sustainable business practices; small and medium sized enterprises

## 1. Introduction

Examining the developments in CSR literature in the past as well as today, antecedent research has recognized a difference in the participation in CSR practices between big companies compared to SMEs [1] (p. 423), [2] (p. 285). According to existing research, CSR has for a long time been considered a topic solely a priority matter for large companies [1] (p. 424), [2] (p. 285). This orientation evolved due to the big and significant impact large firms have on the society and the environment [1] (p. 423). This trend has further been emphasized through technological developments which enable growing visibility of companies' business practices and of their global impact [3] (p. 243). Such developments have accentuated the demand for greater transparency and accountability which is reflected, for example in the general principles and requirements of food law established by the European Union [EU]. These regulations demand traceability of all ingredients of food products as well as of the finished goods, throughout the entire supply chain [4] (p.11). Through such principles, companies—however only large ones—are obliged to officially disclose of CSR related matters [3] (p. 243), [5,6]. A further reason for the focus on large companies is their size and thus the amount of resources available to firms of this magnitude, making them meaningful players with high levels of influence on the CSR topic [3] (p. 242).

Consequently, there is less information available in the field of CSR efforts of SMEs [1] (pp. 423–424). Recent research argues for a re-orientation away from using big multinational companies as a benchmark for CSR [2] (p. 285). The reason is the high level of social and environmental impact of SMEs on most economies due to their sheer number [1] (p. 423), [3] (p. 241). The level of significance of SMEs for CSR is reflected by their high density and thus economic impact. In 2016, within the EU-28 (non-financial business sector), the 23.8 million SMEs accounted for an added-value of 4.030 trillion Euros and the employment of 93 million people [7] (p. 12). Even though as individuals, SMEs might have relatively little impact, taken as a whole their impact becomes much larger and crucial within the sustainability topic [8] (p. 7).

SMEs have however, specific characteristics which greatly differentiate them from large companies [9] (p. 41). The SMEs limited resources—such as budget, revenues or number of employees—are often referred to as a reason why SMEs tend to have limited divisional structures [3] (p. 242), [10] (p. 182). Due to the smaller number of employees, these have to take multiple functions within the organization, which is why SMEs often lack specific specialists [3] (p. 242), [10] (p. 182). Literature claims that within a company, regardless of its size, key players such as managers and a company's owner(s) have a strong influence on the firm's strategy; due to the small size of SMEs the influence of these key players on the company's strategy is more marked [10] (p. 184). Additionally, a limited budget means simpler capital structures and it creates a necessity for SMEs to rely on their networks of personal relations and the firm's reputation to compensate their limited resources [10] (pp. 183–185). External networks provide SMEs with missing knowledge and resources, compensating for their own limited capacity [11] (p. 172). One further characteristic of SMEs, essential for this study, is the relatively limited level of visibility of the companies. SMEs receive significantly less attention from the media compared to large firms, meaning their actions are less exposed to the changes in public opinions and values [10] (p. 185).

In addition to these characteristics, there are further internal as well as external features of SMEs which distinguish them from large firms [10] (pp. 182–186). Small businesses often have a different ownership structure than large firms, since they tend to be owner-operated, meaning the ownership is closely located to the operating units and therefore has a higher level of influence on these units [12] (p. 41). SMEs also only have a few shareholders whilst large companies usually are responsible to a bigger number of shareholders [10] (p. 183).

Furthermore, research has found that the culture of SMEs is significantly different from the culture of large companies, mainly characterized by an informal nature [10] (p. 183), [13] (p. 17). Their culture can be described as more freely structured and intuitive, built on a trusting basis compared to the culture of their larger counterparts [13] (p. 17).

Taken together, considering that the focus of CSR practices has mainly concerned large companies and therefore most reports on CSR relate to large firms, further research on SMEs and their implementation of CSR strategies is still needed [14] (p. 140), [15] (p. 6), [1] (pp. 423–424), [16] (p. 339). Therefore, the purpose of this research is to contribute to this need of additional investigations on the analysis why some SMEs face challenges when integrating CSR practices into their business strategy. Additionally, the development of approaches on how these challenges can be prevented or overcome, is an objective of this research. The data for this qualitative study was collected in 2018 through 12 interviews with managers of SMEs and experts in the field of study. 'The St. Galler Management Model' is referred to throughout this article since this model sets 'management' into context with the 'organization' and the 'environment', emphasizing the interplay of all three dimensions [17] (p. 73). Additionally, the newest version of The St. Galler Management Model sets a strong focus on sustainable business practices through emphasizing the importance of stakeholder engagement [17] (pp. 105–109). The categorization of the different management levels, within the organization dimension, further allows the differentiation and analysis of management challenges which managers and owners of SMEs face—this being aligned with the objective of this research.

To contribute to the scientific debate concerning the challenges faced by SMEs the following research question was posed: Why do some SMEs face challenges on the normative, on the strategic, and on the operative management level when integrating CSR practices into their business strategies, whilst others do not and how can SMEs overcome or avoid these challenges? Furthermore, through researching on how possible challenges on the operative management level, when SMEs implement adapted CSR practices, can be avoided or overcome, this study aims to contribute to the establishment of possible practical implementation solutions for SMEs. The following section will illustrate how these research questions were developed.

## 2. The St. Galler Management Model in Relation to CSR Challenges Faced by SMEs

Many companies have realized that they can create several business advantages by moving beyond the minimum responsibility of the firm, by performing voluntary actions towards social and environmental sustainability [18] (p. 13). The extent of beneficial effects may differ for companies depending on the firm's contributions and efforts towards social and environmental sustainability. Nevertheless, antecedent literature has agreed that an overall benefit can be achieved by firms which implement CSR practices, regardless of their size [19,20] (p. 10). According to a survey from McKinsey [19] for example improving operational efficiency, lowering costs and resources as well as strengthening the company's own competitive advantage, are only a few of such reasons why companies engage in sustainability practices. Nevertheless, the main reason why firms pursue sustainable business models is to maintain or improve the corporate reputation, since today's customers set a high focus on sustainability and sustainable business practices [19].

While many of the mentioned benefits of sustainable business practices seem evident and can lead to business advantages for companies, regardless of their size, there are also several challenges which especially SMEs face when focusing on CSR efforts [19,20] (p. 10). The previously described SME specific characteristics pose challenges for SMEs with respect to the manner with which they have treated CSR in the past [3] (p. 243). The objective of the following sections is to identify these challenges and investigate whether there are ways how SMEs can overcome them. Since the objective of this research is to identify CSR challenges from the management perspective of SMEs, 'The St. Galler Management Model' is applied, see Figure 1. The model puts management into context of the organization and the environment, enabling the creation of an integral view on management and its interplay with these other two dimensions [17] (p. 30).

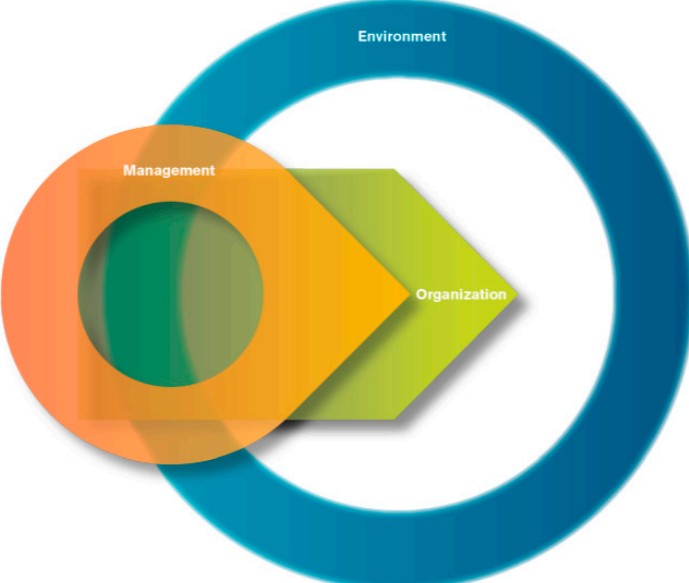

**Figure 1.** 'The St. Galler Management Model'. Source: adapted from [17] (p. 17).

Since the specific ways how SMEs attempt to tackle their CSR efforts differ within these dimensions, the following paragraphs will relate the model to the diverse challenges faced by SMEs. The focus of this research lies on the organization dimension, since on this level the actions of the management are integrated into the management practices, and hence have an impact [17] (p. 197). Therefore, challenges occurring within this process are most efficiently tackled on this dimension.

### 2.1. The Organization Dimension and Its Management Levels

Organizations are potentially value-adding systems, in which decisions, communication flows and actions simultaneously take place. An organization is embedded in a dynamic environment which continuously creates opportunities for the organization, but which also establishes expectations towards the company [17] (p. 118). To enable the organization to establish a coherent and reliable value-added chain, a common reference frame is necessary [17] (p. 178). This reference frame can be divided into three managements levels; a normative level, a strategic level and an operative level, see Figure 2. This frame functions as a guide and legitimation of decisions, communication flows and actions and emphasizes that the tasks of the management vary on each level [17] (p. 178). At the normative management level, a company's culture is established, hence its objectives, norms and values are set. Decisions made on the normative level provide the framework as well as the legitimation for any other decisions made on the lower two levels, which is why the normative management level is placed visually above the other two levels in Figure 2. On the strategic management level, which generates the link between the other two levels, the management focuses on developing business plans and the company's strategy. Here, competitive advantage has to be established and ensured, to secure the firms future. The daily business, however, takes place on the lowest level; the operative management level. At this level the focus lies on an efficient use of resources and on managing the daily operative business in a correct and efficient manner [17] (p. 180).

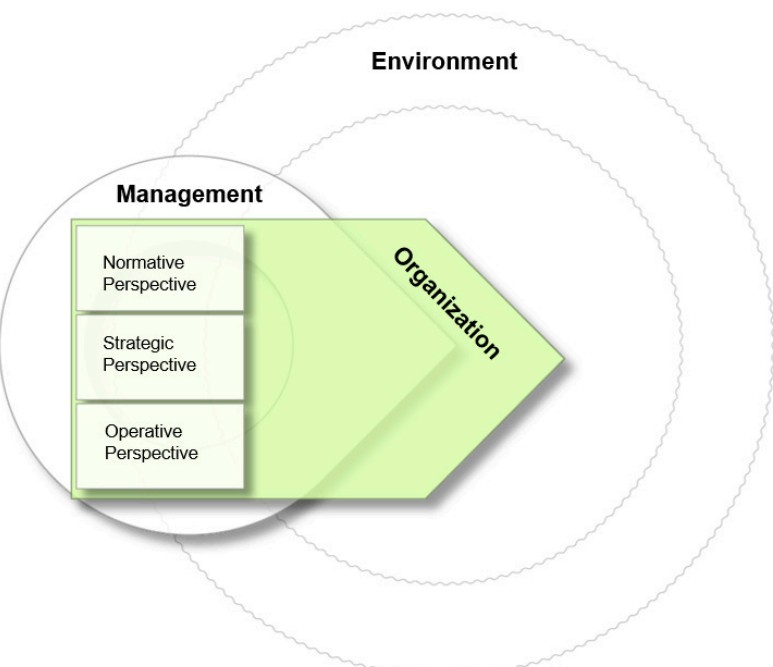

**Figure 2.** The 3 management levels of the organization dimension. Source: adapted from [17].

### 2.2. CSR Challenges at the Normative Management Level

The first level of the organization dimension is the normative management level, where fundamental long-term commitments, existential norms and values, which create the basis for the company's culture are set [21]. Moreover, the firm's identity and its responsibilities are defined as

well as prioritized [17] (p. 181). Hence, it is the task of the management at this level to establish the company's culture as well as to define its relationship and responsibilities towards society and the environment [17] (p. 183).

As has been illustrated, the culture of SMEs differs from the culture of large companies [13] (p. 17), [10] (p. 183). Characterized by a more informal nature, SMEs are often described as more freely structured and intuitive, built on a trusting basis compared to the culture of their larger counterparts [13] (p. 17). The St. Galler Management Model claims that decisions made on the normative management level provide the framework as well as the legitimation for decisions made on the other, lower two management levels [21]. Hence, the informal culture of SMEs trickles down through the other two levels, influencing decisions and practices made there, as will be seen in the following. This indicates the importance of CSR objectives being anchored in the culture of a company.

To achieve a CSR mindset being anchored in the culture of a firm, it has to be reflected in the norms, values and objectives of the company which are set by the management and the owners [17] (p. 183). Research shows that whether companies implement CSR practices is highly dependent on the basic motivation of the owners and managers to do so [13] (p. 492), [22] (p. 27). Ethical-social values are referred to as the main motives of SMEs when pursuing CSR efforts [23] (pp. 27–28), These motives are mentioned more frequently than the goal of increasing the firm's reputation or customer loyalty, i.e., than objectives which focus on the firm's turnover [23] (p. 27). The pursuit of ethical-social values could on the one hand derive from the entrepreneurial status which many SMEs have and their desire to 'return a favor to society' [23] (p. 492). On the other hand, this internal drive to 'put something back into the community' can, in some cases, be linked to religious beliefs of the managers or owners [24] (p. 103). Such ethical-social values seem to be, even within the category of SMEs, related to the size of the company. The smaller the company the more relevant these considerations appear [23] (p. 27). Figure 3 provides an overview of the different challenges which the managers and the owners of SMEs face on the normative management level. Based on the identification of these challenges, the first research question aims to identify why some SMEs face challenges on the normative management level when integrating CSR practices into their business strategies, whilst others do not as well as to how SMEs can overcome or avoid such challenges.

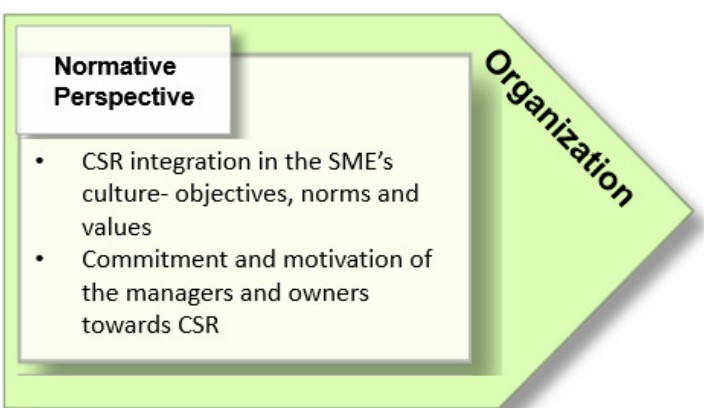

**Figure 3.** Challenges on the normative management level. Source: adapted from [17].

*2.3. CSR Challenges at the Strategic Management Level*

Having presented the challenges which managers and owners face on the normative management level, this paragraph concentrates on the 'strategic management level'. The strategic management level focuses on establishing the company's strategy and ensuring its competitive ad-vantage [17] (p. 180). Through processes and decisions made on this level, strategies evolve which enable the company's long-term survival and success, such as for example the allocation of resources [17] (p. 183). Features specific to SMEs also influence CSR efforts at this management level.

Antecedent research agrees that SMEs are less inclined to implement formal measures and procedures when pursuing ethical business practices, which can not only be recognized in their preferred personal relationships to stakeholders, but also in their preferred focus on dialogue strategies to foster such practices rather than on the use of official instruments such as code of conducts and certificates, which are often implemented by larger companies [25] (p. 45), [16]. This preference can be traced to the less formal culture and structure of SMEs [13] (p. 25). The focus on informal measures and procedures does not create the demand to establish a formal CSR strategy [20] (p. 13). Furthermore, due to their informal and intuitive nature, SMEs tend to not internally nor externally communicate their CSR efforts, compared to the formal and publicly released CSR strategies of larger corporations [20] (p. 13).

An additional strategic challenge which SMEs face when focusing on CSR efforts are their limited resources. These force SMEs to take in a more reactive approach towards CSR practices compared to their bigger counterparts [26] (pp. 270–272). Research has identified a clear connection between a company's financial situation and its level of CSR involvement [13] (p. 491). However, these resources are not exclusively of financial nature, but they also include knowledge, people and time. These restrictions lead SMEs to set their priorities differently, hence challenges closer to home, such as for example employee motivation, customer retention and community involvement gain a higher priority [3] (pp. 243–249), [26] (p. 272). Their limited resources are one of the most often quoted hurdles for SMEs when implementing CSR strategies [26] (p. 272).

The ownership structure also has an influence on the strategic management level. SMEs tend to be, as previously mentioned, owner-operated and literature claims that the top management and owners of such firms find it hard to justify investments in CSR practices [10] (p. 183). This is based on the difficulty for the management to measure such practices and the challenge to identify benefits for the company from an economic perspective, especially in the short-term [26] (p. 270), [3] (p. 247). This shows that regardless of the values and norms of the management and owners, the legitimation of CSR practices represents a big challenge, given that the main goal of any company is to be profitable [27] (n.p.). Especially for newly established companies and smaller firms, the survival and growth of the business has priority, leaving non-commercial goals as secondary [24] (p. 103). Such situational variables have a strong influence on ethical considerations of managers, weighing up the priorities according to the status of business, irrespective of the managers' ethics [28] (p. 126).

Taken together, the characteristics of SMEs generally have had a great influence on how these companies deal with the CSR topic on the strategic management level [17] (pp. 183–186). An overview of the CSR challenges on the strategic management level is provided in Figure 4.

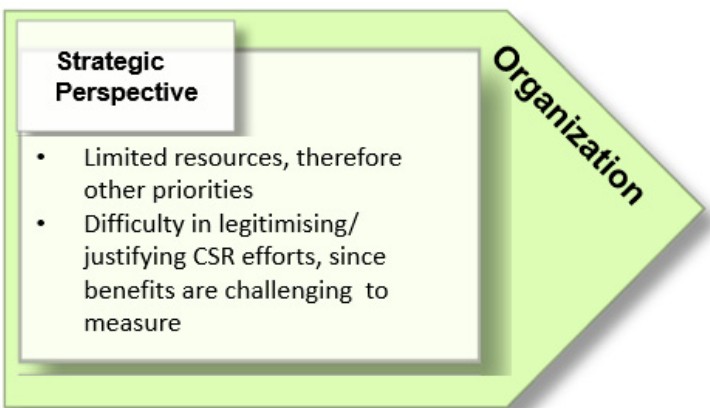

**Figure 4.** Challenges on the strategic management level. Source: adapted from [17].

However, examining more recent literature, research argues that the CSR concept is generic in its nature and is therefore applicable to all companies no matter their size, ownership structure or

industry [14] (p. 142). Furthermore, firms which pursue a focused strategy and thus concentrate their efforts on a single sustainability dimension at a time, seem more profitable than firms investing in multiple dimensions simultaneously [29] (pp. 1281–1282), [30]. It could therefore be argued to focus only on a single CSR dimension at once, makes it easier for SMEs-with their limited resources-to become involved in CSR. These contradictory perspectives from literature emphasize the need to answer the second research question which aims to identify the challenges which SMEs face on the strategic management level when integrating CSR practices into their business strategies, whilst others do not and to research how SMEs can overcome or avoid these challenges.

### 2.4. CSR Challenges at the Operative Management Level

The illustrated frameworks from the upper two management levels both have a strong influence on the third level, namely the operative management level [21]. The focus of this level lies on performing the daily business of the SME as correctly and efficiently as possible [17] (pp. 186–188). Therefore, an efficient coordination of practices and allocation of resources, such as finance, knowledge, experience and time is key [17] (pp. 186–188).

The importance of how CSR strategies are implemented was recognized approximately with the beginning of the new millennium, when research shifted its focus moving beyond notions of 'what' CSR initiatives should be focused upon by companies, to 'how' these should be implemented [29] (p. 1295). Tang et al. [29] (p. 1276) in their studies emphasize that by concentrating on the 'how', given a firms external and internal constraints, the benefits of its CSR efforts can be maximized. However, conventional business cases and theories of CSR are based on the assumption that large companies are the norm [12] (pp. 38–40). Therefore, many CSR approaches have been molded towards big companies, assuming that these cases can simply be scaled down to fit SMEs [12] (pp. 38–40). Such downscaling attempts usually underly certain beliefs which may not be applicable to the average SME [12] (p. 40). SMEs for example often have one, large stakeholder, rather than a wide range of customers, to which they are financially tied to [12] (p. 44). Therefore, the bargaining power lies with the single stakeholder, influencing the nature of the stakeholder relationship [12] (p. 44). Hence, even though the customer relationship may not drastically differ between SMEs and their larger counterparts, the management of these relationships is likely to [12] (p. 44). Considering such cultural differences as well as the specific characteristics of SMEs, it becomes clear that these standardized CSR strategies are too complex and suboptimal for SMEs [12] (pp. 38–40), [14] (p.143).

A research program was conducted between 1995 and 1997 which included 358 SMEs with the objective of identifying challenges which SMEs face when implementing CSR related standards, such as for example the EcoManagement and Audit Scheme (EMAS) [31] (p. 56). The research found that the main challenges which SMEs faced in the operative management when implementing the EMAS standards were linked to barriers from the strategic management level such as limited financial, human and technical resources, but also to issues on the normative management level such as challenges in defining the goals and programs of the environmental management systems [31] (pp. 59–60). Challenges occurring in the implementation process, meaning on the operative management level, were a lack of environmental management skills, in other words not knowing how to interpret and apply the EMAS standards [31] (p. 59). Another study focusing on challenges which SMEs face when pursuing the International Organization for Standardization (ISO) 14001, found that SMEs struggle to adapt these standards to internal processes as well as to manage the required documentation of such standards [32] (p. 732).

Challenges at the operative management level, summarized in Figure 5, underline that CSR standards may still be too complex for SMEs to implement [31] (p. 59), [32] (p. 727). Nevertheless, according to the European Union, even before the publishing of the adapted implementation processes of both standards, the standardized ISO and EMAS were already being implemented by some SMEs [32] (p. 726). This contradiction leads to the question why some SMEs face challenges on the operative

management level when integrating CSR practices into their business strategies, whilst others do not and how SMEs can overcome or avoid these challenges.

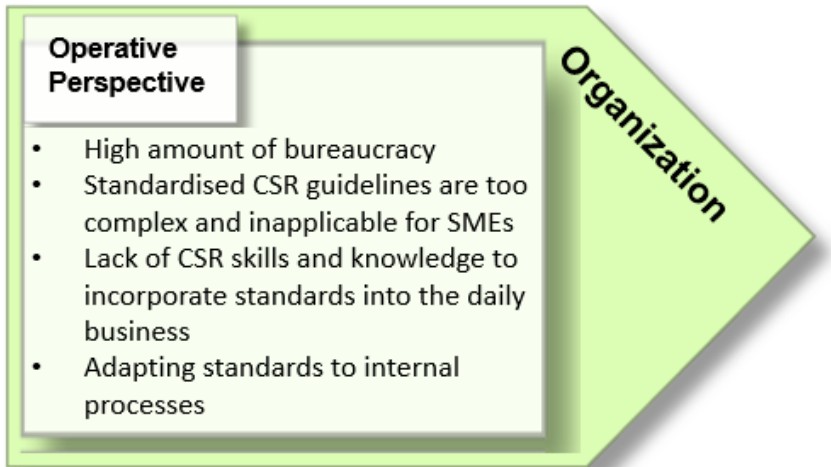

**Figure 5.** Challenges on the operative management level. Source: adapted from [17].

Having recognized the challenges which SMEs face when pursuing standardized CSR guidelines, literature suggests adapted approaches for SMEs which take account of their particular features [20] (p. 5), [10] (pp. 186–193), [25] (p. 47). Therefore, the EMAS as well as the ISO 14001 were adapted to better fit the specific features of SMEs, with the objective to encourage them to integrate such systems [33] (p. 3), [34] (p. 8). Even though the different challenges for SMEs can be traced both to the normative and the strategic management levels, the adapted standards primarily aim to support SME in the implementation process, thereby eliminating issues on the operative management level [33] (p. 3), [34] (p. 8). Furthermore, since the normative management level focuses on the culture of SMEs and on how it prioritizes and defines its social and environmental responsibilities, it can be claimed that the issues on this management level are not relevant to the implementation of adapted or standardized CSR practices [17] (p. 181). Similarly, the degree to which the adapted CSR standards influence the strategic management level is unclear, since the focus of the adapted standards lies on improving the implementation process i.e., the operative management level.

Considering that several guidelines have been established to support SMEs in the implementation process of CSR practices, the question arises as to why there are still many SMEs which have not implemented such practices. Whilst much research has been conducted on challenges which SMEs face when implementing non-adapted standards, the challenges which SMEs face—on the operative management level—when applying adapted CSR practices have, to the best of the researcher's knowledge, not yet been investigated [31] (p. 55), [32] (p. 726). To try to fill this research gap, this study aims to answer the final research question; How can possible challenges on the operative management level, when SMEs implement adapted CSR practices, be avoided or overcome?

## 3. Research Methodology

### 3.1. Research Approach

Even though a large body of research and literature has been established focusing on the sustainability topic, several reports emphasize the need for further research explicitly focusing on SMEs and their CSR efforts [14] (p. 140), [15] (p. 6), [1] (pp. 423–424). Since SMEs tend to be owner-operated, meaning the ownership is closely located to the operating units and therefore the personal values and norms of a manager have a high influence on the business model of the company, this favors the choice of a qualitative research approach for this study [12] (p. 41). Through a qualitative approach the collection of insights into personal experiences and subjective perceptions of the managers and owners

of SMEs is enabled, which is important to enlighten the challenges of the normative management level which focuses on the norms and values of the management. Nevertheless, it must be stated that findings from qualitative research commonly face the criticism of not being representative, since they reflect the personal opinions and experiences of the participants, rather than the reality in its entirety [35] (n.p.). However, literature on research claims that small samples are satisfactory if the objective of the case study approach is to collect data on individual situations and experiences, which is the case in this research [36] (p. 53). Furthermore, potential findings do not have to occur more than once to be included in the analysis [35] (n.p.).

The in literature identified challenges which SMEs face when integrating CSR practices into their business strategies, provide the basis for the practical part of this research, thus a deductive approach is applied. Most existing literature focuses on SMEs challenges when implementing CSR standards originally established to fit the features of large companies. However, no research has been conducted on what challenges SMEs face when implementing the adapted CSR practices. Based on already existing knowledge and findings, the researcher attempts to fill this gap through practical insight from the perspective of SMEs' managers and owners. This approach further underlines the choice of the deductive research approach.

## 3.2. Research Design

Furthermore, this study pursues an explanatory case study analysis. The choice of the case study design is based on the multiple perspectives that this method incorporates as well as being applied on an organizational level of analysis, which is important to understand the studied SMEs [37] (pp. 28–29), [38] (p. 76). Additionally, a case study design with multiple cases, these being the SMEs in different nations, is applied. Due to the similarity of the SMEs in their CSR strategy and achievements, similar findings from the different cases are expected, meaning a literal replication logic was used in the choice of cases [37] (p. 57). This case study design was chosen since a case study design with multiple cases is considered more conclusive and its findings more powerful and robust compared to a case study design with one single case [37] (p. 64). The cases of this study design-the SMEs-are embedded in a shared context, which is the food industry in the DACH-region, meaning in the countries Germany, Austria and Switzerland. Through this shared context, a real-world perspective of the SMEs in the food industry can be achieved [37] (p. 135). Moreover, each of the cases includes three embedded units of analysis, which are the three management levels in the organizational dimension of the St. Galler Management Model and the challenges for the management which occur on each level. This study therefore pursues an embedded case study design [37] (p. 62).

## 3.3. Case Selection and Sample Description

The shared context, being the food industry in the DACH-region is chosen since this industry is not only highly linked to the environment, but it is also characterized by a high degree of visibility and volatility to external demands and pressures from, for example, consumers [39] (p. 307), [40] (pp. 35–36). Hence, the CSR topic is one of high relevance and presence in this industry [41] (p. 1). Furthermore, SMEs accounted for approximately 48.3% of the turnover generated through food and drink in the EU in 2017 as well as for 62.1% of the total employment within this sector [42]. Additionally, antecedent research has found that there is a negative relationship between a company size and their profitability in highly competitive industries [43] (p. 71). Thus, due to either superior products or lower production costs, small companies seem to reach higher productivity levels in the food industry, which is defined as a highly competitive industry due to the strong competition among food processors and high retailer concentration [43], (p. 71), [44] (p. 741). Taken together, due to the high importance of CSR for the companies as well as the high density and success of SMEs within this sector, the choice of the food industry seems suitable for this research topic.

The choice of focusing on Europe is based on the high attention which CSR enjoys in Europe which is apparent through the sheer amount of continuously published CSR regulations in this region.

However, whether such legal regulations influence the manner with which companies in this region approach CSR is not studied. The cases are all companies from the DACH-region as well as all experts work in one of the three countries of this area. The choice of focusing on the DACH-region is based on several arguments. Firstly, within the overall worldwide ranking of the most sustainable countries, all three are listed within the top 20 (Switzerland No. 5, Germany No. 13 and Austria No. 15) [45]. Secondly, since within the food industry ecological efforts are seen to be the most relevant amongst all CSR practices, when the country ranking of the 'Environmental Performance Index', a measure for the ecological performance of a state's policies, is considered [39] (p. 307)—all three countries are listed amongst the top 10 positions worldwide (Switzerland No. 1, Germany No. 6 and Austria No. 8) [46]. Finally, the food industry plays an important role in each of the three countries, accounting for several percentages of each country's GDP. Nevertheless, all these three countries share similar values throughout all the six cultural dimensions of Hofstede [47]. Whether the findings of this study are applicable to firms in countries which are culturally different from those in the DACH-region, therefore represents a further research limitation.

Since an objective of this study is to discover how challenges when integrating CSR practices into a company's business strategy can be avoided or overcome, the focus lies on SMEs which successfully pursue CSR efforts. Therefore, the choice of the specific companies within the DACH-region is based on SMEs with best practices in this area and which recognize sustainability as part of their business identity. According to Dyllick and Muff [48] a truly sustainable business shifts its perspective away from tackling negative impacts of its business practices to "how it can create a significant positive impact in critical and relevant areas for society and the planet" with the resources and competencies which it has at its disposal (pp. 165–166). Hence, SMEs were contacted requesting an interview, if they make "business sense" of social and ecological challenges, by translating these into business opportunities [48] (p. 166). The chosen case companies all have CSR certificates and pursue sustainability standards which indicate their ability to overcome such challenges. Additionally, since this research targets the management of SMEs, the interview partners from all cases represent managers. However, within this sample of SMEs as well as for the choice of experts in the field a haphazard rather than a theoretical sampling strategy was used [36] (pp. 56–57). Reasoning for this is to get insights from the cases in their continuously changing environment, thus creating a real-world perspective [49] (p. 720). By integrating diverse experts with varied backgrounds and thus different areas of expertise, a broad perspective on the topic is hoped to be gained as well as to avoid narrowing down the research focus and possibly missing novel insights. This choice is also based on the need for further research on SMEs and their efforts covering sustainability, due to the high impact that these companies have in this field [14] (p. 140), [15] (p. 6), [1] (pp. 423–424).

### 3.4. Data Collection and Analysis

The practical data from this study evolved from a total of 12 semi-structured interviews which were conducted, in 2018, with managers from SMEs (5) and with experts in the field of study (7) (see Appendix A). The choice of including a higher number of experts in the field is based on their engagement with multiple SMEs, allowing their data to cover a broader SME portfolio. The objective of this chosen method of data collection is to gain personal insights into the perspectives and experiences of the case managers and of the experts, without the scope restrictions which structured interviews entail. All the twelve interviews were conducted personally, the majority in face-to-face meetings and the remaining via phone calls. Disadvantages of semi-structured interviews are its characteristic of being time and resource intense and its difficulties linked to the analysis of the collected data [50]. Nevertheless, the advantages such as flexibility and richness of the data outweigh the negative aspects, which is why this data collection method was chosen [50]. The interviews were scheduled in advance and their length varied between half an hour to an hour. Whilst the majority of interviews were conducted in the English language some were held in German, and directly afterwards translated into English by the researcher. The interviews were recorded, since this allowed the researcher to devote

full attention to the interviewee and to simplify the transcription [38] (p. 166). Two questionnaires were produced, one specifically for the case companies and the other for the experts in the field, which both can be found in Appendix B. While the questions in both surveys are divided into categories and the key questions are included in both, it must be emphasized that the questionnaires served as a guidance for the researcher during the interview and therefore not all questions were necessarily addressed. Additionally, the interview questionnaires helped minimize the common mentioned limitation of qualitative research, namely the risk of biases distorting the data and findings, for example through response biases occurring during an interview due to socially desired behavior [35] (n.p.), [37] (pp. 115–117).

Finally, to ensure that the findings from the collected data provide validity and reliability, several aspects were considered in the data collection of this study [37] (pp. 45–49). To guarantee the consistency and strength of the findings, data should have multiple sources of evidence as well as several methods for the data collection. Through the collection of secondary data from literature reviews as well as primary data through interviews, data and method triangulation are assured, which strengthens the construct validity of the study [37] (pp. 120–121). Construct validity refers to the insurance that the correct operational measures are studied for the specific concept applied.

The data analysis is based on theoretical triangulation to ensure and strengthen the internal and external validity of the research [37] (pp. 120–121). Since multiple theoretical concepts and findings i.e., the challenges which SMEs face, are referred to and applied in the data analysis of this study, theoretical triangulation is assured. Consequently, the causal relationship between the outcome of the study and the examined aspects is reinforced. Finally, to ensure reliability of the findings a research process should be structured in a way that the same results can be achieved through a repetition of the research [37] (pp. 45–46). Therefore, after having transcribed all the interviews, the collected data was coded with the analytical codes which evolved through emerged themes and categorized from theory [51] (p. 193) (see Appendix C). The data was coded as complete, meaning the entire information. In this manner, the interview transcripts were examined for further topics of interest, which could support answering the research questions [52] (p. 206). These newly established codes caused the researcher to review previously coded interviews, to ensure no recently discovered concept was overlooked [51] (pp. 195–197). For this coding process the researcher used the computer assisted qualitative data analysis software Atlas.ti. After this had been done, the researcher applied a pattern matching technique, in which theoretical predicted patterns, such as the SME specific features causing certain challenges, were compared to empirically based patterns, these being any patterns discovered through practical insights from the interviews. The use of the pattern-matching method supports the organization and interpretation of the collected data and enables the researcher to better analyze the findings [37] (p. 143).

### 3.5. Ethical Considerations

To safeguard the research being conducted without any ethical issues arising, this paragraph examines ethical considerations as well as the researcher's role throughout the research process. Respect, competence, responsibility and integrity are the four key principles which Braun and Clarke (pp. 61–62) suggest to be ethical issues. To ensure that no ethical issues evolved, the researcher focused upon these four principles throughout the entire research process as well as on the main objective of a researcher of any qualitative case study, which is to do no harm [52] (p. 62), [37] (p. 78). Further requirements of ethical practice are, for example, to ensure that the collected data from interviews can be treated confidentially and the interview partners remain anonymous [52] (p. 63). Therefore, all the interviewees were given the choice, directly before the interview, to decide based on a 'Disclosure Consent Form' firstly, whether the interview could be recorded and secondly, how their provided data was allowed to be used. Through such ethical practices it was ensured that the interview partners were motivated to openly and honestly share information. Whether, however, the statements of any of the study participants are biased based on their role or function in the field of study, cannot be judged by

the researchers and therefore represents another limitation of this study. Finally, honesty and accuracy were also applied in the data analysis and in the reporting of the findings [51] (p. 63).

## 4. Results

This article continues with the empirical presentation and analysis of the findings from the research. The aim of the following subsections is to create a holistic and meaningful understanding of the in practice identified patterns. To enable a coherent structure of the data presentation and analysis of the empirical findings, each of the three levels of the organization dimension from the St. Galler Management Model will be applied. Due to the dynamics of the different levels, it is well possible that patterns are not strictly tied to a specific level, so that overlaps may occur.

### 4.1. The Organization Dimension

#### 4.1.1. The Normative Management Level

The collected data depicts a strong pattern between the culture of a company, the commitment and motivation of the manager or owner of a firm and the defined responsibilities of a company. However, interviewees are not consistent on whether the specific features of SMEs—such as their low hierarchy and small size-have an influence on how sustainability is integrated into a company's culture. Whilst some stated that there is no causal relationship between SMEs features and their CSR commitment nor success, others claimed that the specific characteristics of SMEs present an advantage for the company when integrating CSR into the firm's values and norms. Nevertheless, the reason mentioned most often as to why some SMEs are successful in their CSR practices is the high commitment and involvement of the top management or the owner towards sustainability. If the manager is committed towards sustainability, he will set the tonality and objectives accordingly. Several interviewees stated that the motivation and commitment of the management determines the direction and path which the company will pursue. Several case interviewees emphasized that within their company the owner or managers strongly stand behind sustainability and that they "really have it within their hearts" (SME 2). The execution of sustainability values from the top management levels was often quoted as a necessity for such values to be anchored within a company's culture and to become an integral part of the firm's DNA. Moreover, according to the collected data, the execution of such values and a commitment of the top management towards sustainability, reflects the importance CSR takes in within a company. The motivation of the managers seems to have a connection to the amount of resources allocated to CSR topics. Managers which are highly committed towards sustainability are determined to create resources for this topic, despite the fact that efforts towards sustainability and the invested resources do not achieve recognizable short-term financial returns. Taken together, the identified pattern claims that if the top management or owner of a company is committed towards sustainability and commits resources to this topic, then a culture will evolve which is dedicated towards sustainability.

Many interviewees mentioned that once anchored in the culture of a company, CSR values permeate through the entire organization, reaching every hierarchy level. In this respect, several interviewees said that in order to ensure that employees have the company mindset concerning sustainability, new employees are either obliged to attend courses in which they learn and discuss about the company's values including those of sustainability or they are handed the company's sustainability report to familiarize themselves with the company's CSR values.

Another pattern observed within the data is that the interview participants, especially the consultants, emphasized that to ensure sustainability becoming a part of the company's identity, CSR not only has to be anchored within the culture, but also in the firm's business strategy. Through integrating sustainability within a firm's strategy and structure, sustainable business practices can evolve.

That the sector and legal framework as well as the external attention which a firm receives, may affect the commitment of a manager towards implementing CSR business practices, was widely acknowledged amongst the interviewees. For example, due to the high relevance of sustainability

within the food industry and the legal regulations of this sector within the DACH-region, managers in this industry are more likely to be willing to commit to and invest in CSR, compared to managers from other industries and regions. This phenomenon increases if the firm receives external attention from for example the media or from NGOs. This is the case within Switzerland which has "a very active NGO-sector" (Expert 2).

A further repeatedly mentioned pattern connected to the normative management level and which seems to impact a company's success concerning its CSR practices, is the history and pioneering status of the company in connection with the SME's culture. Established companies open to new opportunities were more open to familiarizing themselves with CSR, which strengthened the integration of sustainability into the company's values and hence culture.

Taken together a multitude of patterns on the normative management level emerged in practice; several of which reflect the interdependence of the three management levels. The following paragraph continues with the presentation of the data on the strategic management level.

### 4.1.2. The Strategic Management Level

The company interviewees often mentioned that their company's history, not only enables sustainability to be strongly anchored in the firm's culture, but also has a marked impact on the firm's strategies. The many years during which sustainability has been a key factor in their company strategies has allowed the SMEs to develop the CSR topic, enabling them today to take in a "role model" (SME 3) function within their field, and thereby creating a competitive advantage. Whether the company's history has a direct effect on the SMEs strategy or whether its impact is more indirect through the company's culture was not explained nor mentioned by the interviewees.

This competitive advantage appears to be essential, since the most frequently mentioned aim of SME's strategy is to follow a path that differentiates them from their competitors. This is reflected within the collected data through the strong connection between the strategy of a company and the SME specific characteristics. Due to these SME specific characteristics, such as their smaller size, their limited resources or overall impact, the majority of the interviewees argued that SMEs are forced to find strategic approaches through which they can differentiate themselves from their bigger counterparts. As an SME manager summarized "We are a really small 'fish', meaning we cannot compete head-to-head with the biggest players in this branch" (SME 4). Therefore, SMEs strategically search for and focus on means which create a domain for the company, enabling it to differentiate from bigger counterparts. Often mentioned were the creation of own product labels and brands, but also the application of specific standards or by setting the focus on niche markets, or on products and projects which have a unique characteristic.

One point often raised was that a firm's competitive advantage, related to its strategy, and SME specific characteristics, needs to be maintained as a company grows. The majority of the company interviewees asserted that their firms are growing and in due course any SME-specific advantage might become lost. A suggested measure necessary to overcome this challenge, is having sustainability values integrated within the SME's culture and that these values continue to be prioritized by the top management. As a SME manager emphasized "[ . . . ] if [ . . . ] we start growing [ . . . ] it's really very important that, especially the leaders, have this idea of sustainability, that they live these values and the idea." (SME 2). Another measure cited within the case companies how to overcome this potential issue is the necessity for (especially growing) SMEs to report on their sustainability efforts. A regular communication of sustainability topics was stated to be an important step for a SME's strategy to be successful. While many different communication tools were mentioned, such as for example a social media tool for the internal use (SME 2), sustainability reports were frequently referred to as a necessity for SMEs which are growing. Since the creation of a report requires resources such as time and people who are especially employed for this, several interviewees agreed that small companies often lack such resources and therefore their emphasis related especially to 'growing' SMEs.

Some interviewees claimed that the main purpose of a sustainability report was to serve as a company-internal tool to measure the firm's developments and to support team-building. One of the case SMEs stated for example that through the establishment of their report their "employees became prouder and the social cohesion became stronger" (SME 3). Other interview partners stated that the main target audience of such reports were rather players outside the company, such as stakeholders, the public and customers. An expert in the field, who shared the latter view, proposed a reasoning for why sustainability reports in general are established. Within the standardized balance sheet, the sustainability efforts and achievements of a company cannot be distinguished. However, sustainability reports allow a more differentiated view and thus enable the stockholders to recognize the additional efforts of a company. This issue of not being able to identify the output of a company's CSR efforts on a balance sheet, and thus the necessity for specific reports was mentioned by numerous interviewees.

Another distinct pattern which emerged relates to a firm's strategy and its strategic measures. Along with the importance of strategic reporting, the creation by management of specific actions, which should be aligned with the firm's main objectives of sustainability, was frequently referred to as key to ensure a successful CSR strategy. These strategic measures can take in several forms such as for example KPIs, Balanced Scorecards or award systems. The interviewees explained that by establishing such measures a company can trace its performance and can strategically steer the firm to ensure that the set CSR goals are achieved. The experts emphasized that a lack of these measures of performance is a main reason why company's fail to successfully execute their defined objectives. Another mentioned reason for this failure is that the objectives must be set and backed up by the top management, otherwise they will lose attention and thus effectiveness. This is explained by the statement of one of the interviewed experts (Expert 6) that today "many companies don't see sustainability as a strategic issue", which is why the creation of such strategic measures focusing upon CSR is not considered by many managers. However, the problem most often mentioned is that the company fails to break down the CSR strategy into goals which effectuate tasks for all involved parties in all hierarchical levels and which are being measured and controlled.

### 4.1.3. The Operative Management Level

According to the participants, CSR challenges on the operative management level do not automatically evolve but are rather created through the manner with which the CSR strategy of a company is executed in its business practices. Several experts as well as company interviewees agreed that whether the execution of the sustainability strategy of a firm involves, for example, a high amount of bureaucracy, highly depends on how its strategy is being lived within the company. When considering how this CSR challenge could be overcome or avoided, a broader pattern appears which includes not only the commitment of the top management but also the specific characteristics of SMEs. A SME manager for example claimed that due to its company's characteristics such as its flexibility and loose structure, such challenges as high bureaucracy do simply not exist (SME 2). Further, if the managers and the owner of a company live the sustainability values and align the company's objectives to these, such challenges should be, according to several interviewees, avoided. Nevertheless, the data from the company interviewees shows a connection between the business practices of SMEs and challenges concerning CSR. The majority of the companies did mention a challenge they continuously face is to ensure that the focus on the SME's sustainability objectives is not lost in the hassle of the daily business. As another manager of a SME stated "All the employees have so much to do throughout the day, so I think it's difficult not to lose focus on sustainability thought." (SME 2).

Having illustrated two relationships that became apparent at the operative management level, the following paragraphs focus on practical findings concerning the CSR related standards. The collected data indicates that in practice the choice of whether to implement a standard is defined as a strategic decision, meaning a decision which is made by the management and which is usually, according to the majority of interviewees, based on the financial profit the company can achieve through the implementation of the standard. An expert in the field emphasized that figuring out "the reason 'why'

is crucial" (Expert 5), when analyzing companies which implement CSR standards. Interview findings suggest that companies will only implement standards if they can achieve an added value from doing so. Two sources of added value for a company when implementing a standard were identified. Firstly, it can lead to an improvement in the firm's management system, since standards help the management of a company analyze key areas which are essential for the firm. In addition, they support the management in optimizing company procedures, developing more integrated processes. Secondly, if there is an external demand for a specific standard, the company will implement this standard to satisfy this demand and hence create an added value. Many experts did not argue that the high costs are the main barrier why some companies do not implement standards such as ISO or EMAS, but rather that these were seen as "unnecessary investments" (Expert 5), if no added value could be perceived. Hence, according to the interviewees, if no added value can be achieved through the standard, many companies do not see a reason why they should apply such standards. In regard to these standards it should here be mentioned, that the interviewees hardly ever distinguish between normal standards and adapted standards, nor did they differentiate between SMEs and their larger counterparts. The achievement of an added value was given as the main reason why companies implement standards, regardless of the company's size and whether these were adapted or non-adapted standards.

Having illustrated the findings on the reasons why companies implement standards, several findings reflect that the implementation of standards is challenging for a SME. A repeatedly mentioned challenge faced by SMEs, which is also linked to their specific features, is the lack of an employee exclusively focusing on developing and strengthening CSR within the SME. "The problem was not that we had to change the mindset or integrate it, this was already here. [ . . . ] We were lacking an employee that focused on the sustainability topic and on its communication [ . . . ]." (SME 3). Case SMEs which were focusing on implementing standards, did not only face the challenge of lacking a sustainability manager, but also of missing knowledge of the documentation process required to implement such standards. One of the SME managers explained "[ . . . ] we had to learn the process of such documentations. We have been living these values and practices for many years, but all of a sudden, they have to be documented, meaning written down" (SME 5). However, through working together with other SMEs, through external support of consultants or through hiring an employee to focus on the CSR topic, the interviewed companies were able to overcome this challenge.

A further interview finding, which falls into the pattern illustrated above, is that many companies fail to analyze which aspects of the standards are really relevant for their businesses. To focus on specific areas within the standards seems to represent a challenge for companies. The interviewed CSR consultants stated that this was often a task for which they were employed by companies.

Having described the cited challenges SMEs face when implementing standardized guidelines, patterns and perceptions from interviews concerning the adapted implementation processes of standards will be discussed. Even though several interviewee companies are certified according to either the ISO 14001 or EMAS, none had applied any of the adapted standard versions, the reasons for this being varied. A connection between standards and the company's history was one. One manager explained that for years their company had been working with non-adapted standards and in relation to the creation of the adapted implementation processes: "Since we already had a well-functioning system, which was well established and integrated at the time [ . . . ], we didn't think it was necessary to switch." (SME 4). Moreover, a frequently mentioned reason was that the employees had, over the years, learnt and familiarized themselves with the requirements and processes of the standards, which is why the adapted implementation processes are deemed unnecessary.

Another reason why adapted standards were not implemented is apparent from the connection within the collected data between standards, SMEs specific characteristics and soft outputs of a firm's business practices. The findings from the strategic management level show that SMEs put much emphasis on means whereby they can differentiate themselves from competitors. According to many company interviewees SMEs preferably focus on their own systems, brands and logos, thus on so called "soft outputs", to achieve added value. These soft outputs, create value for SMEs which is greater than

that from having a certified standard, since these own systems and brands enable the SME to differentiate itself from others. Furthermore, whilst many experts did not regard the costs of a certified standard of being too high, several of the representatives of the companies clearly stated that this was the case. Not only the high costs, but also the high efforts required for standards have an inhibiting effect on many of these companies. Due to these barriers and their own value adding systems, if there is no external demand for a specific standard SMEs do not see any sense in implementing them. A manager when referring to the ISO 9000 stated, "This all costs, the company has an effort also due to the annual audits and so far, there is no demand for this standard. It currently just doesn't make sense [ ... ]" (SME 5).

In general, the usefulness of these adapted standards was critically reflected upon by the experts. This assertion was justified by the experts through the explanation that the objective of a standard is to help a company achieve better integrated processes. However, whether this originates from an adapted or from a non-adapted standard, was claimed to make little difference. An expert mentioned that solely applying a standard does not mean the company's processes are automatically improved (Expert 3)—any benefit comes from what a company creates out of standards, rather than the format the standards take. While the overall opinion was that academically such adapted implementation processes make sense, in practice, however, it was claimed that they have limited value.

Nevertheless, some opinions were also heard expressing the view that such adapted implementation processes were of value for newcomers in the sustainability topic, serving as "an initial approach or access for beginners" (Expert 6). As a basic tool, which enables the companies to progressively learn through gradual steps with the objective to thereafter focus on the non-adapted standards, these adapted standards can be of use. None of the experts stated whether the supporting function of the adapted implementation processes were solely for SME-beginners, or whether big corporations which are new to the CSR topic, could also benefit from these adapted standards.

## 5. Discussion of Findings of the Organization Dimension

This section aims to combine the findings from practice with those from literature, with the objective of answering the established research questions of this research.

### 5.1. The Normative Management Level

With respect to challenges from literature which SMEs face on the normative management level, both theory and practice claim that a key reason why some SMEs are so successful in their CSR practices is the high commitment and involvement of their top management and/or owner towards sustainability [23] (pp. 27, 492).

Even though many company interviewees acknowledged to have faced several of the challenges described in the literature, having a motivated and committed management was referred to as a main reason why the SME was able to overcome such challenges. Table 1 presents an overview of a few statements from experts in the field of study which exemplify this connection.

The analysis of the collected data from the interviews does not give any indication that this explanation as to why some SMEs face literature-quoted challenges or why some are able to overcome them, would differ on the three different management levels within the organization dimension. Therefore, the findings from this study conclude that having a committed and involved manager or owner, is crucial to overcome or avoid challenges on the normative, strategic, but also on the operative management level. This is further reflected in the congruent findings of literature and of practice which emphasize that the manager's commitment and involvement influences the company's culture which thereafter waterfalls down through the entire company, and thus is also reflected within the SMEs strategy and business practices [21]. While literature does not offer any advice on how upper management can be motivated towards CSR, several approaches identified in practice on how a company's culture can be influenced were illustrated in Section 4.1.1. 'The normative management level'. Since a finding from the collected data was the high connection between a company's culture and the commitment and motivation of the management, it is assumed that CSR consultant's efforts on

influencing a firm's culture, also have an impact on the latter. The most effective channel to influence the commitment of managers and owners is according to the findings, nevertheless, by talking to each other and discussing CSR topics. Additionally, many interviewees claimed that today, and especially within the food industry, sustainability is such a hot and important topic that the management and owners cannot avoid considering CSR. Also, due to the legal restrictions and the external attention related to sustainability which companies in this sector receive, lack of management or owner commitment to CSR was seen by the majority of the interviewees as an impossibility.

**Table 1.** Examples of expert statements regarding the normative management model. Source: own table.

| Interviewee | Statement |
| --- | --- |
| Expert 1 | Regarding the challenges which many of the case companies claimed to have faced, expert 1 emphasized that "by having an intrinsic motivation to integrate sustainability topics into the management system, into the culture and whatever [ . . . ] If that is the case, they will not avoid the difficulties, but they are willing to tackle them" |
| Expert 2 | Expert 2 claimed that challenges, such as the limited resources of SMEs or the lack of CSR skills and knowledge within the company, "are only functions of how someone within the organization, namely the owner or the management, perceives the relevance of sustainability". |
| Expert 6 | "The commitment has to be there from the top management level. [ . . . ]. This differentiates successful from not successful companies; it has to be established at the top levels. Furthermore, I think it is really important that thereafter it is lived." |
| Expert 7 | "The manager or owner has to be determined and committed to hold on to sustainability practices, even when the balance sheet and the income statement speak another language." |

Regarding the first research question, why some SMEs face challenges on the normative management level when integrating CSR practices into their business strategies, whilst others do not and how SMEs can overcome or avoid these challenges, it can be deduced from the discussion of the findings from this research that a prerequisite for the SME is to have a committed and motivated manager or owner. Only with such an asset can the challenges not only on the normative management level, but also on the lower two management levels, be overcome or avoided. It can be assumed that if the top management or owner of a company is highly committed towards sustainability, the necessary resources will be made available and bonded to CSR, which will enable a culture to evolve which is dedicated towards sustainability. Finally, an important success factor for the case companies is their relatively long company-history as well as their firm's past pioneering activities. By embracing new opportunities, they were open to familiarizing themselves with CSR, which reinforced the integration of sustainability into the company's values and hence culture.

Figure 6 presents possible solutions, developed through this research, for the CSR challenges which SMEs face on the normative management level. The next section aims to discuss the findings from theory and practice regarding the challenges which SMEs face on the strategic management level.

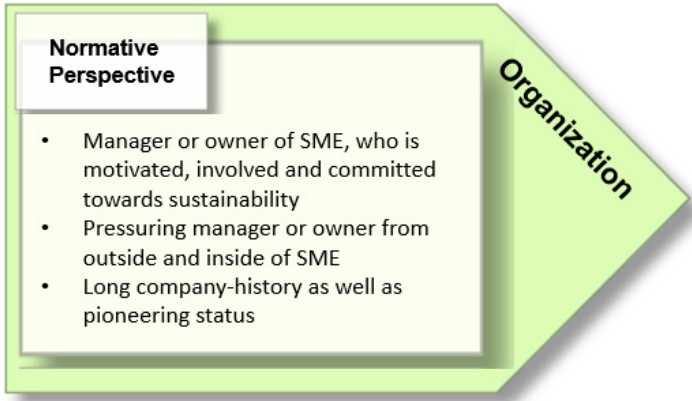

**Figure 6.** Solutions on the normative management level. Source: adapted from [17].

*5.2. The Strategic Management Level*

Considering that antecedent literature asserts that SMEs, due to their limited resources, take in a more reactive approach towards CSR practices and that there is a connection between a firm's financial situation and its level of CSR involvement, the findings from practice claim differently [26] (pp. 270–272). Even though there was no argument from the interviewees that SMEs do have limited resources to create a formal CSR strategy, the specific characteristics of SMEs such as restricted resources are actually one of the most frequently mentioned reasons within the collected data for the SME's success within CSR. Findings from practice claim that due to such features, the SMEs are forced to find strategic approaches to differentiate themselves from other, bigger players within the same industry. These innovative strategic approaches towards sustainability, then appear to create a competitive advantage for the firm within the field. This advantage was often quoted by the company interviewees as a reason why they can avoid the challenges given in the literature with respect to the strategic management level, such as limited resources. Recent literature seems to agree, for example Porter [30] supports the view that small businesses, when compared to their big counterparts, have a high potential with their strategies and limited resources to pursue a clear vision and direction. Due to their small team and the alignment of all players within the organization, upon which the company's strategy is built, SMEs have an advantage compared to larger companies [30]. Additionally, the 'first mover advantage' can be applied to SMEs which implement CSR practices: due to the novelty of the topic for such firms and hence the differentiation of these SMEs from their counterparts, they can achieve a further competitive advantage [53], [33] (p. 71). This is again supported by the practical findings of this study, since the majority of the examined companies mentioned that their long company history enabled them to develop a learning culture, and they thereby take in a "pioneering" approach to sustainability.

A further finding of this study is a concern within the case SMEs that future growth might pose an obstacle to their current strategy and thus competitive advantage, since these are linked to an SME's specific characteristics. To overcome this issue, interviewees emphasized on the one hand, the importance of the sustainability values being anchored in the company culture and being actively supported by top management. On the other hand, a regular communication of their CSR efforts, through for example a sustainability report, can strengthen a firm's CSR strategy and ensure that a growing SME preserves its competitive advantage in this area.

In this respect, a challenge identified in literature which managers and owners of SMEs face on the strategic management level is the difficulty of legitimizing CSR efforts, since benefits evolving from these efforts are difficult to measure [26] (p. 270), [3] (p. 247). The findings from practice indicate that a means to overcome this problem is the publication of sustainability reports. Through such reports a clearer view of the firm's CSR efforts is obtained, thereby providing justification to the top management of their company's sustainability investments. Literature also states that SMEs, based on their informal and intuitive nature, tend not to internally or externally communicate their CSR efforts [20] (p. 13). However, all interviewed companies examined here do apply some kind of communication tool as part of their sustainability efforts and this is seen as a useful tool to legitimize a SME's CSR efforts. It can therefore be assumed that missing to internally and externally communicate CSR efforts is a reason why some SMEs face challenges on the strategic management level.

A final finding from practice, key to overcome or avoid challenges on the strategic management level, is the establishment and institutionalization of strategic measures which are aligned with the SME's CSR strategic objectives. It appears essential that these measures are not only continuously monitored by the top management but also that they are broken down into tasks for all involved employees. Table 2 below presents a few statement examples made by the experts in the field which reflect these discussions.

**Table 2.** Examples of expert statements regarding the strategic management model. Source: own table.

| Interviewee | Statement |
| --- | --- |
| Expert 2 | "If you have a reward system which is not congruent with your goals, you won't get anywhere. Whatever is rewarded, is what will be done in the end." |
| Expert 3 | Only once these strategic measures are successfully established and the tasks implemented, monitored and measured, then "it's a value you can integrate much better into the financial reporting". |
| Expert 6 | Expert 6 while referring to the credibility of a firm stated that this "can only be achieved if CSR is not only being executed internally within the company, but when it is being talked about towards the outside. I personally, think this is the goal." |
| Expert 7 | When talking about the reasoning behind the establishment of sustainability reports, expert 7 elaborated that these reports reflect the "business success, which goes beyond the standard balance sheet". |

Taken together, the practical findings lead to the conclusion that internal and external communication of a SME's CSR efforts, in terms of the establishment of sustainability reports, not only seems to support the SME to sustain its competitive advantage in this field as it grows, but also seems to help the top management and owner legitimize the company's CSR efforts. This conclusion is novel since antecedent literature argues that SMEs do not communicate their CSR efforts. The challenge of SMEs limited resources inhibiting their CSR efforts, as given in the literature, appears not to be entirely correct, since this can be balanced by SME specific features—such as flexibility and sleekness, which, according to practice, can create a competitive advantage over their larger counterparts. Finally, the establishment of strategic measures, which are congruent with the SME's sustainability strategy and which are being controlled and measured by the top management seems key to overcome or avoid challenges on this level.

Suggested solutions how SMEs can overcome or avoid the identified challenges on the strategic management level are presented in Figure 7. Having developed options for the issues which SMEs face on the upper (normative and strategic) management levels, finally the operative management level will be discussed.

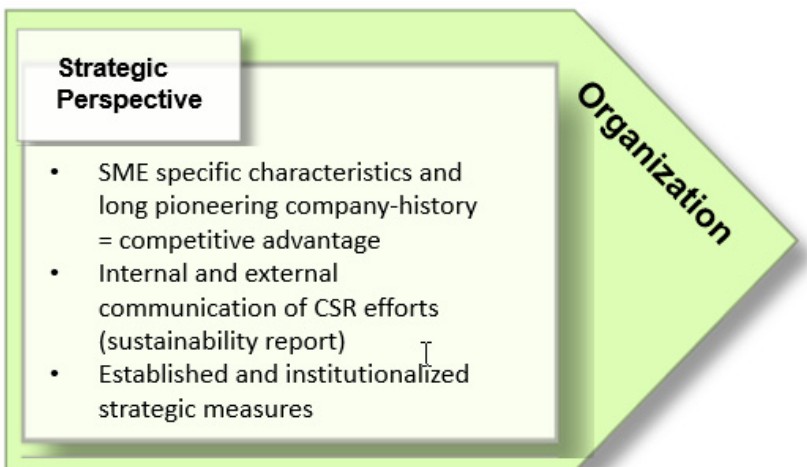

**Figure 7.** Solutions on the strategic management level. Source: adapted from [17].

*5.3. The Operative Management Level*

There is a consensus between theory and literature that the operative management level is highly influenced and shaped by the upper two levels of the organization dimension of The St. Galler Management Model. Therefore, the identified challenges on the operative management level, such as

for example the high amount of bureaucracy, are a result of how CSR is being perceived and lived within the company on the other two levels.

An issue acknowledged by the interviewees which SMEs face, even if they have already implemented CSR into their business practices, is the challenge of not losing focus on the sustainability objectives of the company during the daily hassle. How this challenge can be eased was not clarified by the interviewees.

With respect to the standards, the majority of interviewees agreed with the perception of antecedent research that the standardized CSR guidelines are complex and inapplicable for SMEs, therefore making them difficult to implement [12] (pp. 38–40), [14] (p. 143). The practical findings show, however, that by either collaborating with other SMEs, by engaging external consultants or by hiring or training a company employee to specifically focus on the CSR strategy, the in literature identified challenge of missing skills and knowledge to implement such CRS standards, can be compensated.

The second literature challenge which SMEs face when focusing on implementing standards is the difficulty of adapting standards to their internal business processes [32] (p. 732). Practical findings agree that this process still represents an issue for SMEs. Even though none of the interviewees stated how this challenge could be directly avoided, several did provide advice on how this issue could potentially be eased. SMEs should filter out the most relevant aspects of a standard for their business and set the focus on those features. Moreover, the findings of this study show that through analyzing the stage at which the SME is with its sustainability efforts and by breaking down the future path into achievable steps, successful execution of CSR business practices can be achieved. Table 3 presents several citations from experts in the field which are in line with this conclusion.

**Table 3.** Examples of expert statements regarding the operative management model. Source: own table.

| Interviewee | Statement |
| --- | --- |
| Expert 2 | Expert 2 when referring to standardized CSR guidelines: "If you are a SME your influence, your leverage is limited, hence you don't have to fulfil everything". "Do not play it by the book, play it in a way that it really makes sense for your organization [ . . . ]". |
| Expert 3 | Explaining the reason behind the creation of adapted CSR standards; rating agencies are "mostly focused on selling these to certain institutions." |
| Expert 6 | Regarding how CSR is executed in companies: "It is only bureaucratic if I execute it bureaucratically". Further expert 6 explained that key was to quantify the stage at which the company is with its CSR efforts and to break down the future path into "possible and manageable" steps. |

Concluding it can be stated that literature and practice agree in regard to the explanation why some SMEs face challenges on the operative management level when integrating CSR practices into their business strategies, whilst others do not and how SMEs can overcome or avoid these challenges. Whether a SME faces challenges on the operative management level it seems to highly depends on how sustainability is integrated and lived on the upper two management levels, i.e., on the strategic and on the normative level. What occurs on these two levels will waterfall down onto the operative management level, impacting how CSR is executed in the daily business of the company. Therefore, a successful implementation on the upper levels of the developed solutions of this research should facilitate the same on the lower, operative level.

Additionally, SMEs which try to unilaterally implement standards without necessary knowledge and thereby miss the key focus areas relevant for their business, appear to face the challenges given in the literature. Therefore, by setting a focus on the for a SME's business relevant aspects of standards as well as to filling the knowledge and resource gap—through external support—are, based on the findings of this research, key success factors to help SMEs avoid or overcome challenges on the operative management level. Figure 8 presents a summary of the established solutions to the challenges which SMEs face on the operative management level when integrating CSR into its business practices.

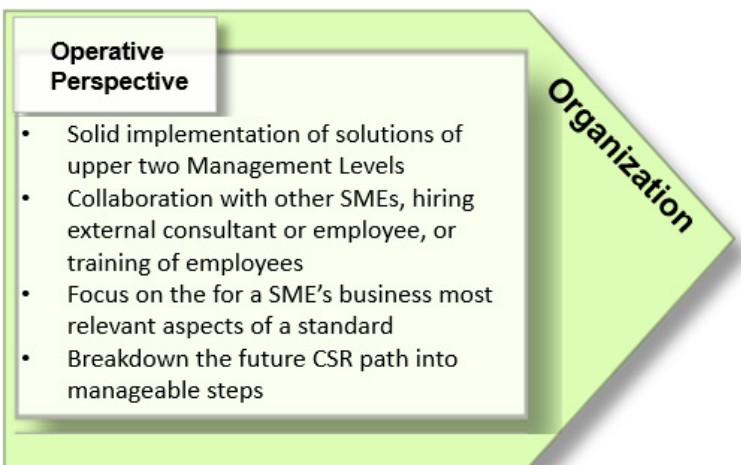

**Figure 8.** Solutions on the operative management level. Source: adapted from [17].

Having addressed the research question focusing in general on the challenges of the normative, the strategic and the operative management levels, the following paragraphs analyze the findings concerning the adapted standards, with the aim to answer final research question.

Since none of the case SMEs had applied an adapted standard version for the implementation process, nor had any of the experts ever worked with any of these, this confirms the literature question—why there are still so many SMEs which, despite the creation of adapted standards, have not yet implemented any of these regulations. Having an adapted standard, does not automatically mean the SME's will apply it, as an expert stated "a booklet for SMEs does not answer the question; 'why' a SME would want to implement a standard" (Expert 5). A finding from practice indicates that a company will only implement a standard if it can perceive an added value from doing so, whether this be through an improvement of their management system or through satisfying an external demand and this is regardless of whether a firm applies an adapted or a standardized implementation process to achieve this additional value. Based on this analysis and on the findings from practice, the usefulness of these for SMEs adapted implementation processes of standards appears to be limited. Moreover, practice has further shown a preference of SMEs to focus on achieving this added value through soft outputs, since these enable the SME to differentiate themselves from competitors.

In conclusion it can be stated that whether there are specific challenges on the operative management level which SMEs face when implementing adapted CSR standards, cannot be answered through the findings of this research. Furthermore, whether these might differ from the challenges identified in literature for the operative management level for SMEs focusing on non-adapted standards, remains an open question.

However, a finding from practice is that in order to design a standard specifically to support SMEs and "that will actually be used, you need to involve SMEs" (Expert 3). The experts were of the opinion that only if the SMEs were involved in the creation of such a standard would it be successful. For example, by defining together with SMEs what kind of data they can and cannot obtain and which qualitative questions have to be asked to gain insights into the company, a standard could evolve which matches the specific needs of SMEs.

All research questions have been addressed and possible solutions were identified as to how SMEs could overcome or avoid challenges when integrating CSR into the company. These developed solutions do not only target the managers and owners of SMEs but all employees—this reflects the importance of CSR being an integral element of the entire company, thus reflected in all aspects of the firm. The following final section provides a conclusion of this research project as well as addresses the research objective of this research.

## 6. Conclusions and Future Research

The particular characteristics of SMEs, such as limited resources, owner-operated management or flat hierarchies, can create challenges for these companies, which also have an influence on the manner with which they address CSR [3] (p. 243). SME's in the food industry are especially challenged with respect to CSR, due to the sectors high degree of visibility and high customer demands regarding this topic [3] (p. 243). Since no business case for SMEs on the implementation of CSR strategies exists, this study set out to identify guidance for SMEs facing the challenge of CSR [20] (p. 5), [10] (pp. 186–193). Furthermore, it aimed to provide input that could assist more SMEs to incorporate CSR practices into their business strategies. The collected data for this study derived from a total of interviews with managers of SMEs as well as with experts in the field of study. These were conducted in 2018.

By combining several reasons why some SMEs face challenges when focusing on sustainable business practices, one identified factor on the normative management level appears to be a root cause for other challenges to evolve, namely a missing committed and motivated manager or owner of the SME. In other words, a prerequisite to overcome any challenges linked to sustainability appears to be that a SME has a committed and motivated top management. This practical finding is in line with antecedent research which claims that whether companies implement CSR practices is highly dependent on the basic motivation of the owners and managers [23] (p. 492), [23] (p. 27). Other difficulties are evident, as illustrated in the literature, but a conclusion of this research is that the chance that these can be overcome is far higher with a committed top management. Additionally, based on the findings of this research the most effective channel how to influence a firm's culture seems to be by talking to each other and by discussing the sustainability topic within the company.

A conclusion of this study concerning the strategic management level, is that the specific features of SMEs can have a positive effect on their CSR success. Their characteristics, such as limited resources, seem to force SMEs to find innovative strategic approaches which thereby differentiate them from their larger counterparts, thereby creating a competitive advantage. In this respect, a challenge for SMEs which was not mentioned in the literature became apparent: due to the high interlinkage between the SME's specific features and its CSR success, many of the case SMEs consider that any competitive advantage of being a small, flexible company could diminish as the company grows. The findings of this research suggest that a means how to overcome or avoid this possibility is to formalize CSR efforts both internally as well as externally, for instance through the publication of sustainability reports. Moreover, issues on this level can be minimized through the establishment of strategic measures that are continuously monitored and measured by the top management and which incorporate tasks for all involved parties within the firm's sustainability efforts.

Finally, concerning the challenges on the operative management level, a conclusion of this research is that the implementation of the above solutions on the upper two management levels-especially ensuring a CSR committed and motivated top management-seems crucial to overcome and avoid challenges on this level.

Concerning the different standards which aim to provide companies with supporting guidelines how to successfully integrate sustainability into their businesses, a conclusion of this study is that companies, regardless of their size or whether these are standardized or adapted guidelines, appear to only apply such standards, if they can achieve either an internal or external added value by doing so. In addition, it was apparent that SMEs generally lack the necessary specific knowledge, skills and resources to be able to implement these standardized guidelines. A proposed solution is to network and collaborate with other SMEs and/or through obtaining external support such as hiring CSR consultants or by educating an internal employee in this field. Another conclusion as to how to ease the difficulty of adapting standards to a SME's internal business processes was to focus only on the most relevant aspects of a standard as well as to break down the future sustainability path into manageable action steps.

A final objective of this research was to fill the current research gap as to whether there are any challenges which SMEs face on the operative management level when focusing on adapted CSR

standards. However, the collected data failed to provide clear insights and as such this gap remains. However, one conclusion from the practical findings is that for such an adapted standard to be successful i.e., actually used by SMEs, they have to be part of the development of the standards, so that their specific needs are taken into account. That the final research question stays open is further justified since the solutions for the challenges which SMEs face when focusing on standardized guidelines are congruent with the objectives which the adapted standards aim to achieve. For instance, the proposal of this research to 'Focus on the for a SME's business most relevant aspects of a standard' is very similar to the declared objectives of the revised version of the ISO 14001 standard, which states that "This guide aims to help organizations [ . . . ] identify areas for improvement." [34] (p. 8). Additionally, this research's solution 'Breakdown the future CSR path into manageable steps' is congruent with aim of the established manual for SMEs by the GRI to "help SMEs take the first small steps in their sustainability journey" [54]. Hence, if the adapted standards incorporate the previously defined solutions for the challenges which SMEs face on the operative management level when applying non-adapted standards, the requirement for further research on this topic might not be necessary.

The findings and conclusions of this research identify areas of interest for future research. For instance, considering the overall diversity of the functions and roles of the interview partners within the sustainability topic, their statements and perspectives do not, however, reflect this diversity. Main areas of focus and challenges were, with little exceptions, commonly agreed upon. Possible explanations for this like-mindedness, could be found within the findings from the management dimension of The St. Galler Management Model. While literature claims that the engagement with other managers and owners, thus with people who share similar functional profiles, leverage the reflexive process of the individuals, the collected data claims for a more differentiated view [17] (pp. 196–203), [13] (p. 25). Exclusively engaging with like-minded people, who share similar functions and are from the same sector network, according to the findings of this research, may not be the optimal way to manage CSR. Therefore, further research could be undertaken as to how to broaden the views within companies working on CSR thereby bringing in a broader spectrum of ideas.

A final aspect which could be of interest for future studies, is the finding that the integration of SMEs in the definition of standards specifically targeting SMEs, is essential for the standards to be successful. Through researching on how SMEs can be integrated in the creation of such standards, the likelihood of their application and thus success could be increased.

**Author Contributions:** Conceptualization, A.C.E. and C.-H.D.; formal analysis, A.C.E.; investigation, A.C.E.; methodology, A.C.E. and C.-H.D.; project administration, A.C.E.; supervision, C.-H.D.; visualization, A.C.E.; writing—original draft, A.C.E.; writing—review & editing, C.-H.D.

**Funding:** This research received no external funding.

**Conflicts of Interest:** The authors declare no conflict of interest.

## Appendix A.

**Table A1.** Overview of Interview Partners.

| Interviewee | Company/Organization | Position | Country | Description |
|:---:|:---:|:---:|:---:|:---:|
| SME 1 | (Anonymous) | Project Manager Sustainability | Switzerland | Chocolate producer, and a subsidiary of a Swiss retailer |
| SME 2 | Jucker Farm AG | Manager Marketing and Communications | Switzerland | An agritourism company and agriculture producer and distributor |
| SME 3 | Holle baby food GmbH | Sustainability and Marketing Manager | Switzerland | A manufacturer of organic baby food |
| SME 4 | Kärntnermilch reg. GmbH | Environmental Manager | Austria | Dairy producer |
| SME 5 | Naturata AG | Quality and Environmental Manager | Germany | Manufacturer of biologically produced groceries |
| Expert 1 | Swiss Business Council for Sustainable Development (öbu) | CEO | Switzerland | Swiss network for sustainable economies (>360 members) |
| Expert 2 | BHP—Brugger und Partner AG | Senior Consultant | Switzerland | Consulting company with a core competence in 'Corporate Social Responsibility' |
| Expert 3 | Centre for Corporate Responsibility and Sustainability (CCRS), University Zurich | Director of the CCRS | Switzerland | Associated institute at the University of Zurich, focusing on exploring the role of the private sector in sustainable development on the local and global level, Jury member of the Zürcher Kantonalbank (ZKB)-KMU Award |
| Expert 4 | Eartheffect GmbH | CEO | Switzerland | Company offering workshops, educational programs and consulting on sustainability |
| Expert 5 | E2 Management Consulting AG | CEO | Switzerland | Consulting company, which develops and implements environmental and sustainability management for companies,Former managing director of öbu |
| Expert 6 | (Anonymous) | Project Manager | Austria | Economic platform for CSR and sustainable development, Organizer of a nation-wide CSR award |
| Expert 7 | Regionalwert AG Freiburg | CEO and Founder | Germany | A citizen shareholder company that channels citizens' money to build up regional sustainable enterprises |

**Appendix B. Interview Questionnaires**

*Appendix B.1. Interview Questions for Experts in the Field*

Appendix B.1.1. Introductory Questions

- Why do you think that a few SMEs are successful in their CSR practices whilst others are not?
- How do the characteristics of SMEs influence their success in their CSR practices?

Appendix B.1.2. Challenges on Different Management Levels

　　Short information on usage of "St. Galler Management Model" and division in three levels
　　When integrating CSR practices into a SME's culture, strategy and into its daily business several challenges such as e.g., the following have so far been identified in literature:

- Normative Management Level

　　○　CSR integration into the mindset of the management/owner(s)?
　　○　No evident relation between CSR and the company's activities
　　○　Lack of consideration and understanding of the CSR topic
　　○　CSR integration in the SME's culture

- Strategic Management Level

　　○　Limited resources, therefore other priorities
　　○　Owner-operated, leading to difficulty in legitimizing/justifying CSR efforts due to difficulty in measuring benefits of efforts
　　○　Limited external attention and pressure
　　○　No demand for formal CSR strategy due to e.g., personal relationships with stakeholders

- Operative Management Level

　　○　Lack of CSR skills and knowledge
　　○　High amount of bureaucracy
　　○　Standardized CSR strategies are too complex and inapplicable for SMEs
　　○　Incorporating standards into the daily business, adaption to internal processes

- Why do you think some SMEs face these challenges whilst others do not? (cause of challenges)
- How can SMEs overcome these challenges?
- What is key to avoid these challenges?
- What is the role of the manager/owner of an SME when integrating CSR practices into the culture, strategy and into the daily business of the company?

Appendix B.1.3. Adapted CSR Standards

- How do you judge the development of adapted CSR standards to better suit SMEs characteristics?
- Why do you think there are still so many SMEs which, despite the creation of the adapted CSR standards, do not implement CSR practices?
- Do you think that the challenges which SMEs face on the Normative (challenges they face when implementing CSR practices in the firm's culture) and on the Strategic Management Level (challenges they face when implementing CSR practices in the firm's strategy) differ when focusing on adapted CSR standards compared to non-adapted standards?
- Do you think there are any challenges on the Operative Management Level for SMEs focusing on adapted CSR standards? If so, elaborate which ones and how these could be overcome or avoided.

*Appendix B.2. Interview Questions for SMEs*

Appendix B.2.1. Introductory Questions

- What does sustainability stand for within XYZ?
- Why do you think that XYZ is so successful in its CSR practices?
- Does the fact that XYZ is a SME have any influence on its success in its CSR practices?

Appendix B.2.2. Challenges on Different Management Levels

Short information on usage of "St. Galler Management Model" and division in three levels

- Did XYZ face any challenges integrating CSR practices into the company's

  ○ culture (norms, values and objectives)?
  ○ strategy (strategic measures, procedures, allocation and prioritization of resources, shareholder relationship)?
  ○ daily business?

- Elaborate which ones and their causes.
- How did XYZ overcome these challenges?

If so, how and why was XYZ able to avoid, well-researched challenges, such as for example

- Normative Management Level

  ○ CSR integration into the mindset of the management/owner(s)?
  ○ No evident relation between CSR and the company's activities
  ○ Lack of consideration and understanding of the CSR topic
  ○ CSR integration in the SME's culture

- Strategic Management Level

  ○ Limited resources, therefore other priorities
  ○ Owner-operated, leading to difficulty in legitimizing/justifying CSR efforts due to difficulty in measuring benefits of efforts
  ○ Limited external attention and pressure
  ○ No demand for formal CSR strategy due to e.g., personal relationships with stakeholders

- Operative Management Level

  ○ Lack of CSR skills and knowledge
  ○ High amount of bureaucracy
  ○ Standardized CSR strategies are too complex and inapplicable for SMEs
  ○ Incorporating standards into the daily business, adaption to internal processes

- What is the role of the manager/owner of XYZ when integrating CSR practices into the culture, strategy and into the daily business of the company?

Appendix B.2.3. Adapted CSR Standards

- Why does XYZ implement adapted CSR standards? Why not? . . . only if XYZ implements adapted CSR standards:
- Were there any challenges implementing these adapted CSR standards in the daily business of XYZ?
- How did XYZ overcome these challenges?

**Appendix C.**

| Code (deductiv) | Abbreviation | Definition |
|---|---|---|
| Management Dimension_Reflexiv Funcion and Process | MD_RF&P | Critically reflecting upon the company's value-added activities and future objectives. Recognizing innovation potential of sustainability. |
| Management Dimension_Communities of Practice | MD_CP | Engagement of managers and owner with other people, possibly from outside the company, who have similar functional profiles |
| Organization Dimension_Normative Level_Culture | OD_NL_C | Establishing and institutionalizing a company's culture, hence its objectives, norms |
| Organization Dimension_Normative Level_Responsibilites | OD_NL_R | Defining and prioritizing a firm's responsibilities towards the society and the environment, creating a firm's identity. Active decision to focus on CSR and invest firm's resources in this topic. |
| Organization Dimension_Normative Level_Management | OD_NL_M | Motives such as ethical-social values or financial benefits as main drivers for CSR efforts from manager/owner. Manager/owner commitment and involvement. |
| Organization Dimension_Strategic Level_Strategy | OD_SL_S | Vision and mission of a company, including strategic objectives, decisions, positioning of the firm (competitive advantage). |
| Organization Dimension_Strategic Level_Reporting | OD_SL_R | Decision on how and why a firm's strategy and actions are reported and communicated (internally and externally). This also covers financial reportings. |
| Organization Dimension_Operative Level_Business Practices | OD_OL_BP | Procedures and processes through which the every day work is executed in an efficient and correct manner. |
| Organization Dimension_Operative Level_Standards | OD_OL_S | A set of guidelines (standardized or adapted) which aim to support a company in the systematic implementation of specific objectives. |
| Environment Dimension_Resource Configuration | ED_RC | Stock of facilities through which organizational added value can be achieved, e.g. raw materials, 'bought-in' knowledge e.g. consulting. |
| Environment Dimension_Stakeholder Relations | ED_SR | A party with an interest in the organizaiton and which has a high impact on the organization's access to resources. |
| Environment Dimension_External developments | ED_ED | Trends and attention evolving from outside the company, including for example media attention and public pressure. |
| CSR Challenges | CSR_Ch | Issues and challenges companies face when integrating CSR practices into their business strategy. Issues concerning the CSR topic in general. |
| SME Characteristics | SME_C | Features which are specific for companies within the categories small- and medium sized organizations. |
| Large Company Charactericis | LC_C | Features which are specific for companies within the category large sized organizations. |

**Figure A1.** *Cont.*

| Code (inductiv) | Abbreviation | Definition |
|---|---|---|
| Organization Dimension_Strategic Level_Strategic Measures | OD_SL_SM | Systems and trainings which support implementation and quantification of a firm's strategy, e.g. KPIs, Balanced-Scorecard, sanctions, education, financial funds (subventions, compensations), internal governance. |
| Organization Dimension_Operative Level_Soft Output | OD_OL_SO | Not directly measurable or tangible improvements in the output of a company's CSR efforts such as declarations, code of conducts, labels, brands or certificates. |
| Organization Dimension_Operative Level_Hard Output | OD_OL_HO | Measurable or tangible improvements in the output of a company's CSR efforts e.g. improvement of environmental or social issues, as well as financial outputs. Tripple-bottom line. |
| Environment Dimension_Sector | ED_S | Industry, market (-structure), country in which a company operates in. |
| Environment Dimension_Laws | ED_L | Set of regulations specific to a country or industry. |
| Enviornment Dimension_Network | ED_N | Engaging with not only stakeholders, but with other third parties within the same cluster. Community. (Other third party members do not necessarily have to have similar funcitonal profil) |
| Company_History | C_H | Historic development of the firm; e.g. age of the company and pinoneering status. |

**Figure A1.** Code Book.

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
