# Peer review of "Solutions for SMEs Challenged by CSR: A Multiple Cases Approach in the Food Industry within the DACH-Region"

_sustainability, doi:10.3390/su11174758_

Round 1

Reviewer 1 Report

Dear Authors,

Thank you very much for your paper focusing on the challenges of firms (Large and SMEs) doing business in the food industry (in the DACH region, Switzerland), when it comes to CSR activities. The authors conduct interviews with twelve individuals; five SME manages and seven experts. The main findings of the paper that indicate, that if the managers of SMEs become committed to CSR activities, then this positive spillover might potentially spread to other firms (with support and engagement of other stakeholders). After reading the paper, I acknowledge that some interesting observations are emerging, which are in line with the CSR mainstream literature. Nevertheless, I also have a series of methodological questions related to the qualitative research that has been done, and I also have reservations concerning the generalisation of your findings. Moreover, I think that more studies might be considered in the literature review. Given these large improvements expected, I think that the paper cannot be considered for publication in its current form.

1.      I appreciate that the authors introduce some financial numbers related to the food industry. This is very nice, but why it is placed in a section titled “Research Methodology”? I would encourage authors to create a separate section on specifics of food industry based on the data presented in the current version of the paper and based on the review of the relevant literature. The stream of literature on the food industry is quite rich, so I suggest at least works of Hirsch and Gschwandtner (2013); Hingley et al. (2013), Blažková and Dvouletý (2019).

2.      I must admit that I am not very happy about the way how the authors methodologically conducted the qualitative part of the paper. There are many important aspects of qualitative work, and I do not see them in the current version of the article. Although the authors mention categorisation, I could not find any codes and themes in the text, clear description of the qualitative approach, including sample selection and selection of interviewees. How about theoretical saturation, has it been reached? How long have the interviews taken and how about ethical issues? It is also very common that the authors describe examples of categories in the form of a structured table and the demographic information about interviewees are also, unfortunately, missing (although there is a very nice base for it in Appendix 1).

3.      I think that the author needs to be very cautious (as there are five SMEs represented) in any generalisation. Unfortunately, some attempts to generalise are in the article present. I would welcome a special section on comparing insights from the different cases more in a qualitative way. Insights from managers are welcomed, but they should not lead to general implications. The authors should be very cautious at that. They can be only suggested to be tested by the quantitative framework or they should stay with their specific cases.

4.      Empirically, I still do not understand how you can compare (“quantitative research”) large firms and SMEs as you state throughout the text and also in the abstract. This is methodologically not sound, and I think you can only provide insights from expert and SME managers.

5.      The concluding section is very much generalisation of results, and I believe that qualitative papers or case-study papers should look completely different here.

6.      I also think that there should be more discussions related to the role of “expert opinions” and “SME managers” opinion, as there might always be biases in what “experts” are saying. At any point, this is a limitation worth mentioning.

7.      I would also a welcome a special section limitations.

8.      Finally, I think that the title and the abstract (especially) should reflect the actual research that was done. What have you done? How, when?

Literature

Blažková, I., & Dvouletý, O. (2019). Investigating the differences in entrepreneurial success through the firm-specific factors: Microeconomic evidence from the Czech food industry. Journal of Entrepreneurship in Emerging Economies11(2), 154-176.

Hingley, M., Lindgreen, A., Reast, J., Wiese, A., & Toporowski, W. (2013). CSR failures in food supply chains–an agency perspective. British Food Journal. 115(1), 92-107.

Hirsch, S., & Gschwandtner, A. (2013). Profit persistence in the food industry: evidence from five European countries. European Review of Agricultural Economics40(5), 741-759.

Reviewer 2 Report

Dear Authors,

first, I want to thank you and the editor to give me the possibility to read your interesting paper. The topic is relevant and bearer of important practical implication for the development of CSR in SMEs.

There are some points you could improve to make your contribution adequate for the publication.

1) research methods: you can better explain the criteria of choice of the 12 people for the interviews

2) research methods: it is not clear how you analyzed the data collected by interviews

3) in general, you explained that your study is explorative in nature, but you did not place your work in a clear methodological stream

4) I suggest to distinguish better the results from the discussion. In present version of the paper, some risults are reported but not discussed in the light of previous studies, while other results are reported in discussion section. 

5) Finally, the illustrations have a low quality and are not well readable. I suggest to improve the quality of the illustrations.

A part from these minor, in my opinion, the paper has a good potentiality and can positively contribute to scientific debate.

I wish you all the best for this paper and for your research! 

Reviewer 3 Report

Proofreading required. For example

53 which greatly differ them

69 SMEs

78 comma not semicolon

93 colon not semicolon

112 Change While the only few to While only a few

128 “following chapters”?  Chapters are part of a text.

139 Each management level—normative, strategic, and operative needs to be defined before moving into the Normative Perspective

187 These are not chapters. Refer to each as a level as is in the heading

336 The in antecedent ?

376 Use No or # not no

498 “We are really small ‘fish’

255 I think it would be beneficial to add a paragraph describing some of the “CSR approaches” that  “have been moulded towards big companies, assuming that these cases can simply be scaled to fit SMEs.”  You could use the ones in the reference.

Overall, there is value in the research, but the paper has repetition and long sentences.  For example the paragraph starting at 822 does not have to repeat what has already been said about CSR knowledge at the operational level.  The focus here is the finding.  However, if you go back to everything you have said about the “operative management level,” your findings do not address many of them.

I would suggest a table that shows the responses of the experts since some of them are similar.  I also think the questions asked the experts should be included.  This would make the results much clearer for the reader.  Currently, it is not clear what results are significant because they apply to many of the SME leaders interviewed and which ones are not.

Reviewer 4 Report

In my opinion, the theoretical introduction is too broad from the perspective of the topic of article and should be shortened (by 10-15%) for fragments not related to the SMEs. The autors themselves write "... standardized CSR strategies are too complexed and suboptimal for SMEs..." (p.7).

I suggest not to use the term "correlation" (pp. 13-14 - replace them with another word) if the article does not contain correlation calculations.

Round 2

Reviewer 1 Report

Dear Authors, 

although I suggested rejecting the paper in the last round, I am very positively surprised about the changes you made and about the efforts you put into this significant revision. Most of my comments were addressed in a very satisfactory manner. All I would like to change is to add more concrete information into the abstract, introduction and conclusion about the number of interviews conducted and about the time (year). 

Thank you very much for your work and I wish you good luck in your future research.

Your Reviewer;
